# Simulation and Characterization of Nanostructured Electromagnetic Scatterers for Information Encoding

Petr Drexler [1],*, Dušan Nešpor [1], Radim Kadlec [1], Tomáš Kříž [1] and Alois Nebojsa [2]

1 Department of Theoretical and Experimental Electrical Engineering, Faculty of Electrical Engineering and Communication, Brno University of Technology, 616 00 Brno, Czech Republic
2 Central European Institute of Technology, 612 00 Brno, Czech Republic
* Correspondence: drexler@vut.cz

**Abstract:** Nanostructured scattering arrays for the optical spectral domain can be used as passive tags for information encoding, similarly to the manner in which RFID technology does. Setting up their specific spectral response depends on their geometry and the properties of the building materials. The primary design can be provided by using an analytical calculation procedure that is more straightforward and simpler than a numerical simulation. However, the question arises as to the validity of the results. Both approaches are examined in this article. Complementary scatterer arrays were designed using simplified analytical calculation and by means of numerical modeling. The experimental samples were fabricated by the focused ion beam milling of a gold film on a glass substrate and characterized by a spectroscopic system. The results of the analytical calculations, the numerical simulations, and the experimental measurements were compared. On the basis of the comparison, it was observed that for quick array design, both approaches can be used with satisfactory accuracy. Moreover, the simple numerical model also proved the possibility of the identification of the basic dipole mode splitting. Focused ion beam milling was shown to be suitable for the rapid production of complementary scatterer arrays.

**Keywords:** spectral information encoding; nanofabrication; resonance light scattering; electromagnetic scatterer; ion beam milling

## 1. Introduction

Nanostructured passive scattering arrays composed of tuned elements have recently been receiving significant attention. They can be used in various applications, for example for manipulating unusual light propagation; for the confinement of subwavelength electromagnetic fields; as an interactive device in physical or biomedical sensors; or for spectral encoding of information. The latter possibility exhibits similar advantages to those of chipless Radio Frequency Identification (RFID) technology, but in a new electromagnetic spectral range. These arrays can provide a specific spectral response in scattered light following their primary irradiation by a light source. Moreover, the rapid miniaturization of such tags can be achieved. This can be useful for hiding the tags in special cases.

An important step in the design of passive optical domain tags encoding specific information is setting up their specific spectral response within the scattering effect. The spectral response depends on the dimensions of the scattering element, the configuration of the elements in the array, and on the properties of the materials used for the fabrication of these structures. The aim of this contribution is to present the conclusions obtained in the course of research on the design of passive scattering arrays for the infrared region of the electromagnetic spectrum.

Research into the design, fabrication, characterization, and application of micro- or nanostructured passive devices with an electromagnetic (EM) response is a current topic in electromagnetic research. As mentioned above, the specific properties of such structures

enable them to be used, for example, as the building blocks of metamaterial devices [1,2]; novel optical devices [3]; components in biomedical sensing [4,5]; biochemical active structures [6]; or physical quantity-sensing systems [7,8]. Interesting physical effects can be observed in the course of their examination, such as strong dispersive scattering, the formation of localized surface plasmon polaritons, subwavelength electromagnetic field confinement, and others.

To achieve these effects, specific physical responses can be obtained by appropriately engineering the electric or magnetic coupling to an external electromagnetic field. This is normally accomplished by adjusting the shape, dimensions and material parameters of the structure's components, which are made of metals (individual scatterers) and dielectrics (usually a substrate). When we consider these scattering structures as artificial materials, they can be characterized on the basis of their effective permittivity and effective permeability, which can acquire not only positive, but also negative values. To achieve such specific effective parameters, the resonant behavior of the partial structure's components is used. This leads to the fact that the desired behavior can be achieved within a relatively narrow frequency band, the width of which is also influenced by material losses at high frequencies.

Very often, the whole structure will be composed of multiple identical scatterers with mutual coupling, which also affects the resulting EM response. This will also increase the effective interacting cross-section with respect to the outer source or the detector when physically examining the structure. Further advances in research into the design and application of passive structures comprising an array of resonant scatterers were reported in [9]. The resulting spectral response of the whole structure exhibits multiple peaks, which correspond to the different tunings of each scatterer. Adjusting the composition of the array, which is usually of a planar type, determines the resulting spectral response of the whole array. One example of the exploitation of such structures without the lumped elements [10] is robust passive electromagnetic identification, which is currently evolving in the gigahertz area as chipless RFID [11]. A similar field of application for planar or 3D arrays of resonant elements is in frequency-selective surfaces (FSS). In this case, too, an effort to push the operating frequencies into the terahertz and optical regions of the spectrum can be observed [12,13]. Nevertheless, in the case of FSS, all of the elements of the structures are designed to work on a given unified frequency in order to achieve a common functional frequency response.

Advances in research into multi-element resonant structures have been directed towards achieving their operation in the optical spectrum domain [14]. These advances are dependent on developments in micro- and nano-fabrication techniques, since the resonant frequencies of scatterers are inversely proportional to their dimensions. A promising option is the use of specially engineered structures to encode data in the optical spectral domain, similar to the manner in which chipless RFID tags do [15]. In such cases, the whole structure might consist of a certain number of passive scatterer arrays. Each array will be tuned to a defined frequency that provides data encoding. Typically, two selected tuning frequencies will be used for binary information encoding. Therefore, a set of 10 arrays is able to store 10-bit information.

Possible applications of nanosized scattering structures can be found in the security field. The need for nanofabrication facilities for the production of the given structures, the non-trivial manufacturability of the structures, or a specific read-out of the response, represent unique ways of avoiding falsification.

Aspects of the design, modeling, fabrication, and experimental characterization of such nanoscale structures represent a very broad range of research tasks. A large number of proposed structure geometries, simulation approaches, and fabrication and characterization procedures have been examined. The intent of this paper is to describe the findings of theoretical and experimental examinations of the selected approach in the development of complementary dipole structures with response in the infrared optical band, prepared by the focused ion beam milling [16].

## 2. Scattering Dipole Structures for Infrared Band

A basic dispersive scattering structure can be formed as an array of conductive strips on a dielectric substrate. When interacting with an impinging EM wave, charge oscillations might occur between the arms of the strips. Because of the similarity to a classical antenna effect, the strips can be treated as dipoles. To achieve the characteristic spectral response in the optical spectral domain, the dipoles' dominant dimension (length) must be scaled down [17] to between hundreds and thousands of nanometers. Having dipoles of the appropriate length allows the formation of standing waves on the dipoles, leading to changes in wavelength-selective reflection of the through propagation of EM waves. The basic operating mode of the dipole exhibits the largest wavelength. Higher operating modes occur at multiples of wavelength.

To find the required length of the dipoles and to attain the desired response, we must consider the permittivity of the structure's substrate, as well as the dielectric function of the dipole's metallic material. A general relation for length $L$ calculation for dipole mode is in the following form [18]:

$$L = i \cdot \frac{\lambda}{2n_{\text{eff}}(\lambda)} \; ; i = 1, 2, 3, \dots, N,$$ (1)

where $\lambda$ is the desired wavelength of the dipole mode and $n_{\text{eff}}(\lambda)$ is an effective refractive index of the dipole–substrate structure, which can be calculated as follows:

$$n_{\text{eff}}(\lambda) = \text{Re}\left\{ \sqrt{\frac{\varepsilon_{\text{s}} \cdot \hat{\varepsilon}_{\text{m}}(\lambda)}{\varepsilon_{\text{s}} + \hat{\varepsilon}_{\text{m}}(\lambda)}} \right\},$$ (2)

where $\varepsilon_{\text{s}}$ is the permittivity of the supporting substrate and $\hat{\varepsilon}_{\text{m}}(\lambda)$ is the complex dielectric function of the metal used to fabricate the dipole, respectively. The permittivity of dielectrics commonly used as substrates varies only slightly with wavelength. In contrast, for metals such as gold, both the real and imaginary components of the dielectric function change significantly. However, in the middle infrared band, these variations have no significant impact on the deviation of the calculated dipole length from its expected value based only on a consideration of the permittivity of substrate. A more significant deviation is observable in the near-infrared and visible optical spectra.

In one of our previous works, we experimented with the fabrication and characterization of dipole arrays made of gold, deposited on a silicon substrate [19]. A set of five dipole arrays with different lengths were fabricated by means of electron beam lithography (EBL) and the lift-off technology. The scanning electron beam microscope (SEM) images are shown in Figure 1. The dipole lengths in the arrays were 2.09 µm, 3.09 µm, 4.37 µm, 5.43 µm and 6.3 µm, respectively.

The array samples were subjected to spectroscopic examination of their reflectance and transmittance. Due to the spectral range of the detector used, the lower dipole mode ($i$ = 1) could not be detected. However, a significant response was observable at higher frequencies, which corresponds with modes with $i$ = 5. This initial experiment proved the possibility of producing microstructures with a designed spectral response in the optical region of the EM spectrum.

A significant problem of the above-mentioned EBL and lift-off processing is the strong contamination of samples by carbon from organic precursors, which can reach a very high level. The result of contamination is a rapid decrease in the conductivity of the deposed metallic pattern and a low-quality factor of desired resonant behavior. One way of purifying the deposited pattern is sample annealing, which is a relatively demanding procedure [20].

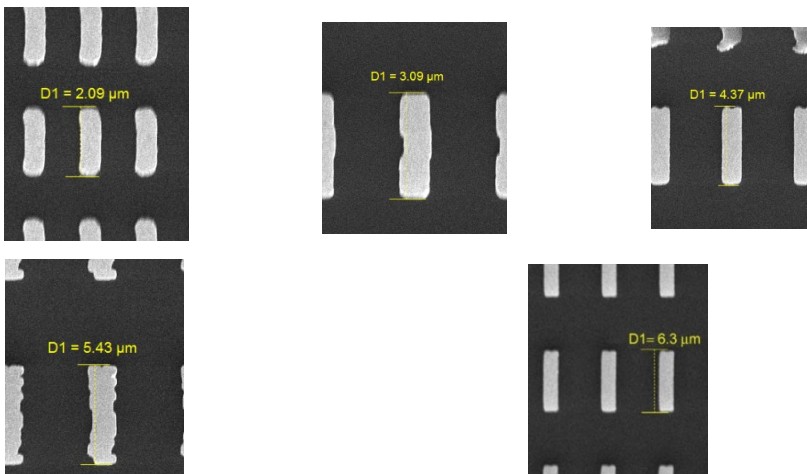

**Figure 1.** SEM images of gold dipole arrays from [19].

To overcome the problem of carbon contamination, we further examined the use of focused ion beam (FIB) milling [21,22] of sputtered metallic film. To reduce the required operation time, the fabrication of a complementary dipole structure [23] was performed. Using this approach, an array of slits in metallic film was produced on a dielectric substrate. With respect to the complementarity of the primary and the complementary scatterer, the frequency position of their characteristic spectral features should be identical.

Optical glass BK7 was proposed as the substrate material. Although it does not allow transmission beyond approximately 2.7 µm, it can provide interesting results with respect to the reflectance characterization of the samples. Furthermore, gold was chosen as the material for the conductive pattern.

Considering the wavelength range of the spectroscopic measurement system intended for further experimental characterization, four dipole lengths were proposed—800 nm, 1200 nm, 2200 nm and 3800 nm. Table 1 presents calculations of the expected wavelength values for basic dipole modes of a single scatterer with length $L$. The second column of the table presents the refractive index $n_s$ of the BK7 substrate, which was calculated by means of Sellmeier equation extrapolation [24]. The third column contains substrate permittivity $\varepsilon_s$ as the second power of the refractive index $n_s$. The fourth column contains the dielectric function of gold $\hat{\varepsilon}_m$ according to [25]. The fifth column contains the calculated effective refractive index pertaining to a gold scatterer on a BK7 substrate. It is obvious that these values are negligible when compared to the original refractive index of the substrate, especially at longer dipole lengths. The sixth column presents the expected values of wavelength at which the basic dipole modes of the scatterers should be observed. The values of $\lambda_1$ in Table 1 were calculated using Formula (1).

**Table 1.** Calculation of wavelength values for basic dipole modes of the complementary scatterers.

| $L$ (nm) | BK7 | | Au | $n_{\text{eff}}$ (−) | $\lambda_1$ (µm) |
| --- | --- | --- | --- | --- | --- |
| | $n_s$ (−) | $\varepsilon_s$ (F·m$^{-1}$) | $\hat{\varepsilon}_m$ (F·m$^{-1}$) | | |
| 800 | 1.488 | 2.214 | $-271 + 24i$ | 1.494 | 2.4 |
| 1200 | 1.459 | 2.129 | $-613 + 79i$ | 1.462 | 3.5 |
| 2200 | 1.452 | 2.108 | $-1986 + 457i$ | 1.453 | 6.4 |
| 3800 | 1.452 | 2.108 | $-5301 + 2026i$ | 1.452 | 11.0 |

## 3. Numerical Analysis of Complementary Scatterers

The infrared spectral behavior of the complementary structures proposed above was examined by means of numerical analysis using a full-wave finite element method solver (Ansys HFSS). The modeled structure was represented by a single slit in a zero-thickness perfect electrical conductor (PEC) layer. Previously performed analyses indicated that

only a slight wavelength (frequency) shift of the spectral peaks occurred when considering either the PEC or the gold model in the simulation [26]. The width of each slit was one tenth of its length in all cases.

The substrate had the defined fixed permittivity $\varepsilon_s$ of BK7 glass, as reported in Table 1. The thickness of the substrate was 500 nm, which turned out to be sufficient. It was verified that the values of electric and magnetic field strength obtained on the basis of the simulations did not vary in substrates with thicknesses higher than 500 nm. The substrate thicknesses of 500 nm used considerably reduced the computation time, since a lot of nodes were spared within the model meshing.

The single slit model was surrounded by boundary layers that provided periodic boundary conditions for the simulated structure. Therefore, the obtained response corresponds to the periodic array of the dipoles (scatterers). A regular Ansys HFSS wave port element was used as a wave source for generating the EM wave impinging on the examined structure.

After setting up the models, harmonic modal analysis with an excitation power of 1 W was performed, and the scattering parameter S11 was evaluated as a measure of reflectance of the dipole arrays. The results of the numerical analysis are shown in Figure 2.

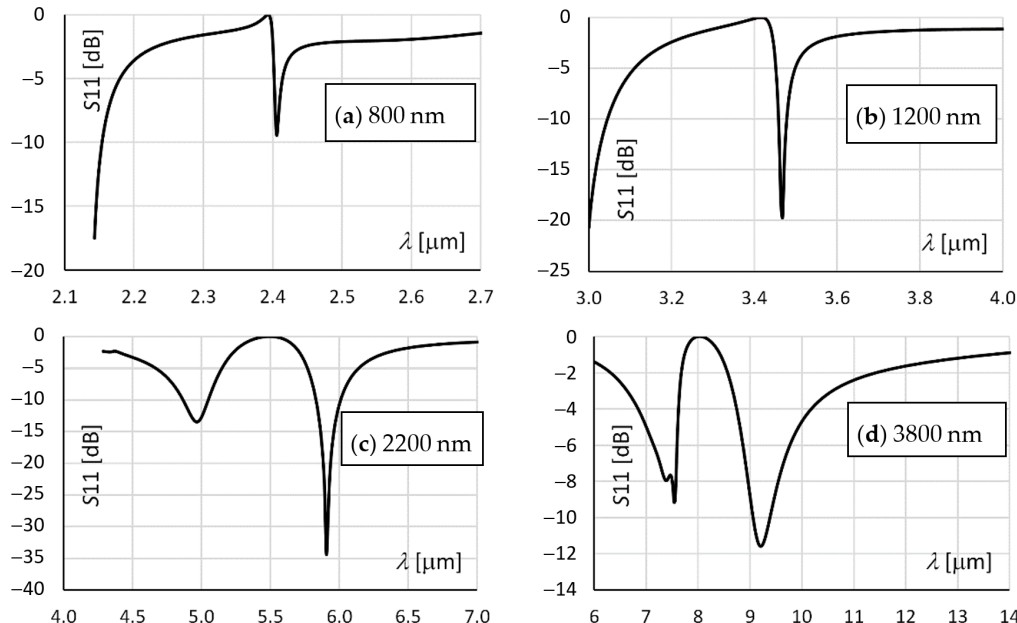

**Figure 2.** Results of numerical analysis of the reflectance of complementary dipole structures with lengths of (**a**) 800 nm; (**b**) 1200 nm; (**c**) 2200 nm; (**d**) 3800 nm.

The remarkable dips in the spectral characteristics of reflectance should correspond to the basic dipole modes. By comparison with the expected values presented in Table 1, it can be seen that there is a good match for dipoles with lengths of 800 nm (2.4 μm calculated, 2.41 μm simulated) and 1200 nm (3.5 μm calculated, 3.5 μm simulated). In the case of dipole lengths of 2200 nm and 3800 nm, two remarkable dips are present in the reflectance characteristic. The dips at longer wavelengths are relatively close to the expected values presented in Table 1: 6.4 μm calculated and 5.9 μm simulated for the 2200 nm dipole; 11.0 μm calculated and 9.2 μm simulated for 3800 nm dipole. The presence of a second dip at a shorter wavelength indicates the possible existence of another resonant mode for the structure. One possible explanation could be that both dips originate from splitting a basic dipole mode, when two cases of impedance matching occur around the central resonant wavelength of the dipole structure. An indication of the other close modes at lower wavelength can also be observed in the case of the 800 nm and 1200 nm dipoles in Figure 2. In all cases, the ratio between the close modes is similarly around 1.2.

Figure 3 presents the spectral characteristics of the electric field strength evaluated at the edge of the slit for the 2200 nm and 3800 nm dipole structures. A significant increase in strength can be observed at the wavelengths of both of the considered modes, evidencing the resonant phenomenon at two wavelengths. However, further investigation will be needed to clarify this phenomenon.

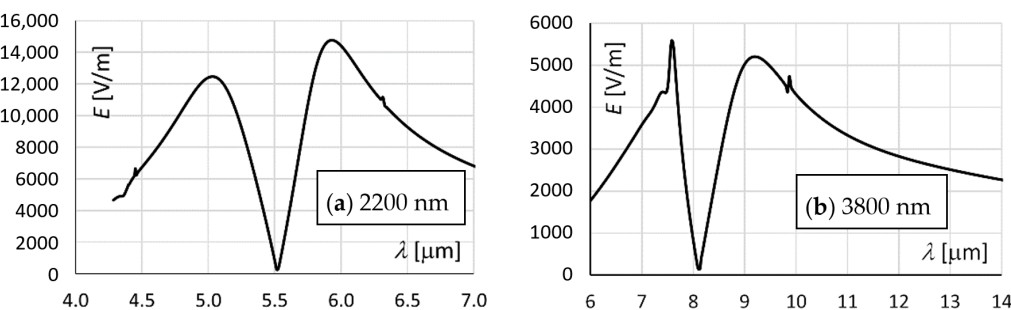

**Figure 3.** Electric field strength on the edge of the (**a**) 2200 nm and (**b**) 3800 nm slit complementary dipole structures.

## 4. Fabrication of Experimental Samples

The proposed complementary dipole arrays were subsequently fabricated to verify their properties. A BK7 plate with a thickness of 0.9 mm was used as a supporting substrate. A tantalum film with a thickness of 3 nm was deposited by means of magnetron sputtering on the substrate as a binding layer. Above that, a gold layer with a thickness of 200 nm was deposited using magnetron sputtering.

The proposed dipole structures were further milled using a gallium-type focused ion beam at a voltage of 30 kV and a current of 6 pA.

Two identical sets of dipole array structures were fabricated, and one of them was inspected by SEM. SEM images of each whole dipole array with its designation and with its detail are shown in Figure 4. The acquired images confirmed the correctness of the fabrication of the FIB-milled structures.

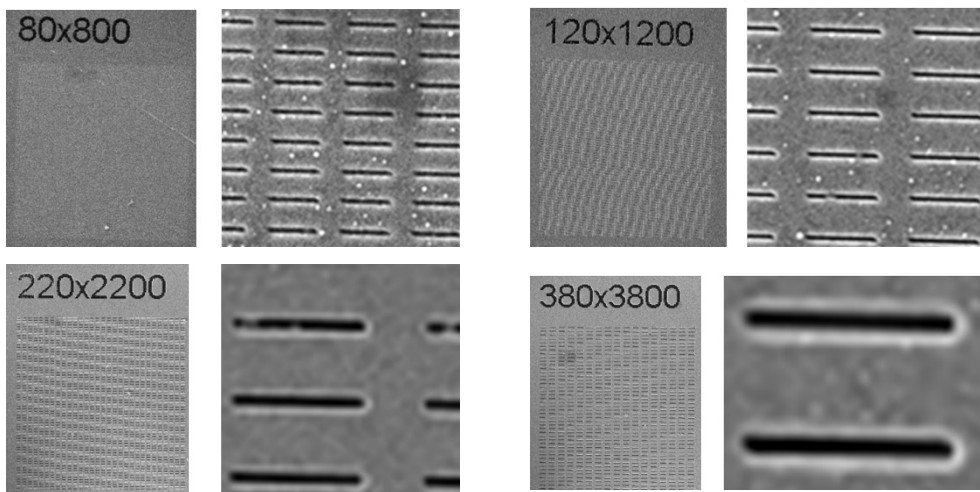

**Figure 4.** Scanning electron microscope images of dipole array structures.

## 5. Characterization of Experimental Samples

The fabricated samples underwent spectroscopic characterization of their reflectance in the mid-infrared band. A Fourier-transform infrared (FTIR) spectrometer, Bruker Vertex 80v, connected with a Bruker Hyperion 3000 microscope and a mercury–cadmium–telluride detector, was used. Characterization was performed using polarized light. The maximum signal magnitude in the response of the two longer dipoles structures (2200 nm, 3800 nm)

was observed when the light polarization was aligned in parallel to the main axis of the dipoles. Meanwhile, in the case of perpendicular orientation, the spectral response was nearly flat, indicating almost no interaction. This observation indicates that the correct dipole mode can be excited in the case of the linear polarization of probing radiation aligned in parallel.

The measured spectral dependencies of the reflectance of the 800 nm and 1200 nm dipole arrays are shown in Figure 5. The results exhibit a nearly flat response with no dispersive behavior for the 800 nm dipole array and a very weak dip at 3.1 μm for the 1200 nm dipole array. Despite this, the SEM investigation of the samples indicated that they had been fabricated correctly.

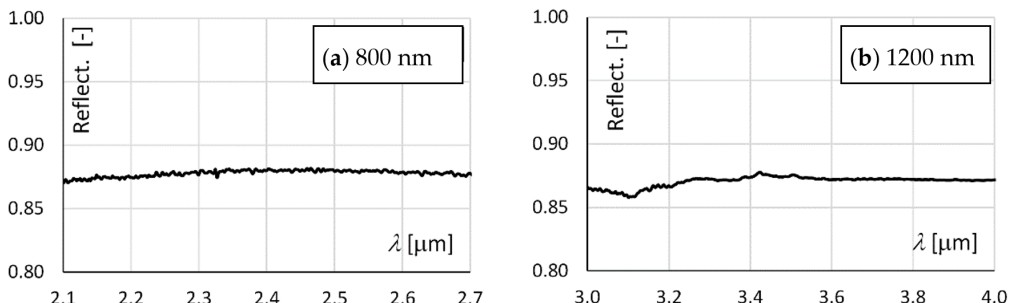

**Figure 5.** Measured reflectance of shorter dipole array structures with lengths (**a**) 800 nm and (**b**) 1200 nm.

To resolve this contradiction in the results, an inspection of the samples was performed via optical microscopy in the visible optical domain. Since the substrate is transparent on visible wavelengths and the dominant dimension of the slits is large, visible light should propagate through the structures and should be observable. Figure 6 presents microscopic images of s the horter dipole structures. The whole arrays and also their details are shown.

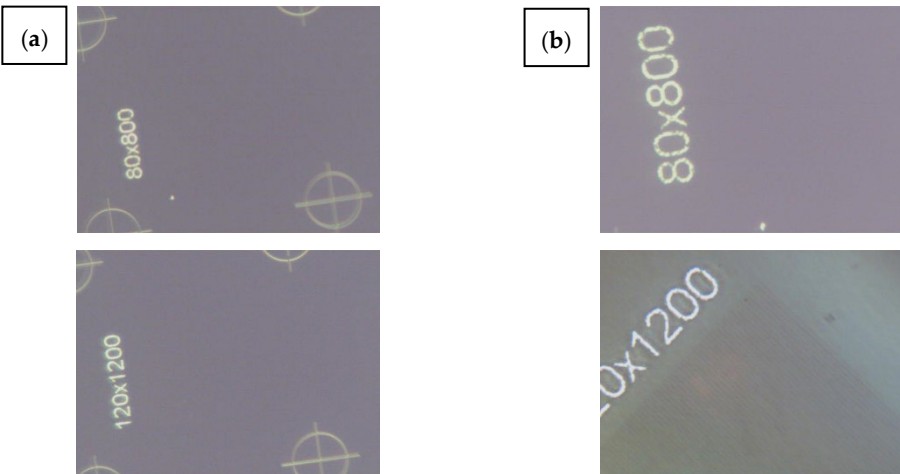

**Figure 6.** Images of short dipole structures in transmittance microscopy at visible wavelengths; whole dipole arrays (**a**); details of dipole arrays (**b**).

A 1:45 microscope objective, together with a 20× ocular, thus giving a total magnification factor of 900, was used to take the image of the whole array, presented as the left part of Figure 6. A 1:100 microscope objective together with a 20× ocular, thus giving a total magnification factor of 2000, was used to take the image of the detail of the array, presented as the right part of Figure 6. It is evident that light propagates through the text designation of the arrays, making them visible. However, the transmission of light through the dipole array is not observable in the case of 800 nm dipole array. In the case of the 1200 nm dispole, a very weak pattern can be recognized. Figure 7 shows images of longer dipole structures

taken with the same microscopy configuration. Here, the array patterns are clearly visible. Our proposed explanation is that during the ion milling of shorter dipole arrays, the slits were to narrow with respect to their depth. This may have resulted in the atoms of gold not being totally removed from the slit, and some amount diffusing into the substrate at the bottom of the metal film.

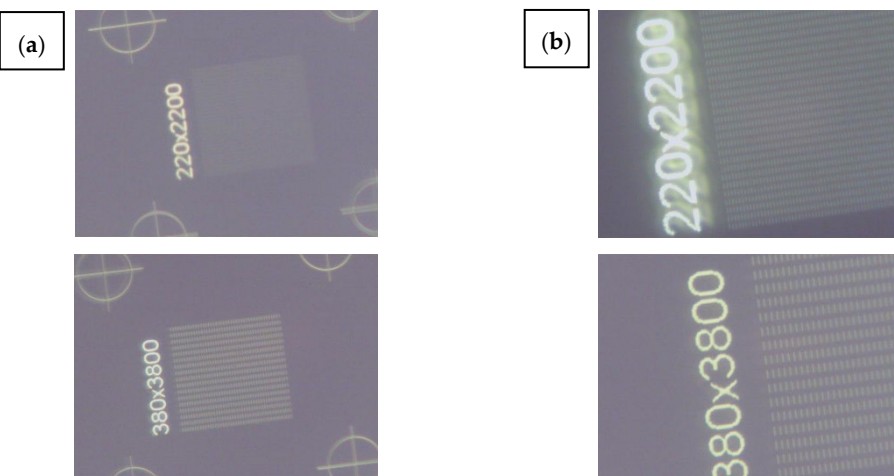

**Figure 7.** Images of longer dipole structures acquired by transmittance microscopy at visible wavelengths; whole dipole arrays (**a**); details of dipole arrays (**b**).

Another cause could be the fact that not all of the material was removed from the slit during milling. This could lead to array opacity, and also to broadband reflectivity, as indicated in Figures 5 and 6. In the case of the 1200 nm dipole arrays, partial weak transparency, as well as a very weak spectrally dependent reflectance could be observed. The weak dip at 3.1 µm probably corresponds to the predicted dip at 3.5 µm.

The measured spectral dependencies of the 2200 nm and 3800 nm dipole arrays reflectance are shown in Figure 8. The results show the spectrally dependent reflectance of the arrays. The 2200 nm dipole array has a very broad dip at 5.8 µm, which corresponds approximately to the analytically calculated value of 6.4 µm and more precisely to the simulated value of 5.9 µm. The 3800 nm dipole array has a broad dip at the higher wavelength of 11.7 µm, which is relatively distant from the simulated value of 9.2 µm, but very close to the analytically calculated value of 11 µm.

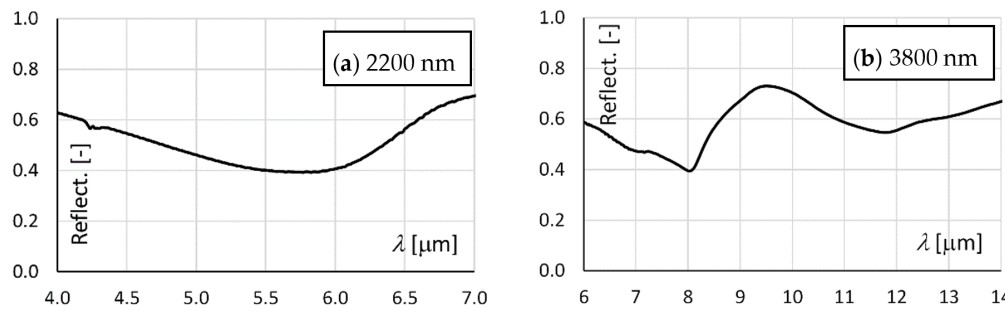

**Figure 8.** Measured reflectance of longer dipole array structures: (**a**) 2200 nm and (**b**) 3800 nm dipoles.

## 6. Discussion of Results

A discussion of the results is provided only for the scattering arrays with dipole lengths of 2200 nm and 3800 nm, since their fabrication was not affected by the diffusion of gold atoms into the supporting substrate during milling. Figure 9 presents a comparison of the simulated S11 scattering parameter and the measured spectral characteristic of reflectance. As mentioned above, the scattering parameter S11 obtained using the EM field simulator corresponds to the reflectance of the structure.

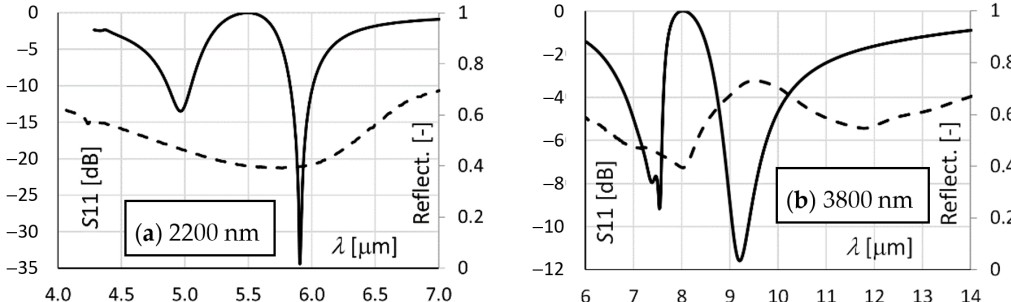

**Figure 9.** Comparison of the spectral characteristics of the simulated scattering parameter S11 (full line) and the measured reflectance (dashed line) of dipoles with lengths of 2200 nm (**a**) and 3800 nm (**b**).

In the case of the 2200 nm dipole, there is an evident decrease in both the S11 and also in the reflectance. This arises very close the analytically predicted wavelength of 6.1 μm. In the case of the 3800 nm dipole length, similar spectral behavior can be observed between the simulated and measured characteristics. Nevertheless, a significant wavelength shift in the identified dipole modes is obvious. This could be caused by certain differences between the geometries of the model and the fabricated slits. Another important fact is that, with a relatively thick layer of gold (200 nm), the inner walls of the slit are not parallel after FIB milling. On the contrary, they subtend by a certain non-zero angle. Therefore, the real geometry is different from the modeled geometry of the slit with parallel inner walls.

The slits with diverging inner walls probably also have broad dips in the spectral characteristics. This is because of the possibility of the formation of many dipole modes, since the dimensions of the slit are continuously varying in the vertical direction. Anyhow, the spectral broadening of the dips is remarkable. The spectral broadening of the dips is probably also influenced by the limited conductivity of the metallic part of the array as a result of residual contamination during sputtering and milling. A secondary consequence might also be the shift in the wavelength of the dipole modes.

The possible influence of the material properties of the substrate also has to be mentioned. The substrate material used was BK7 glass, for which the extinction factor (i.e., the attenuation of light transmission) rises rapidly above 2.7 μm. Above that frequency, the increased losses might also significantly affect the quality factor of the spectral responses and the wavelength shift of the dipole modes. It is likely that this effect came into play, because the standing wave at the metal-substrate interface exhibits significant damping.

Table 2 summarizes the results obtained for the spectral dips of the dipole modes, determined on the basis of spectroscopic measurements of the fabricated samples, by means of simple analytical calculation, and by means of FEM modeling and simulation. When considering the measured values as the reference, it can be concluded that simple analytical calculation considering the refractive index of the substrate and the dielectric function of the metal can be used. The resulting absolute relative deviation $\delta_r$ was under 13%. In the case of the simulation of structures, the deviation was also low. The exception to this was the simulation of the 3800 nm dipole array, where the deviation of the wavelength response was 21.4%. However, this could be rated as a good result, since only a simple simulation model was used.

**Table 2.** Comparison of the results of the examination of the basic dipole modes.

| L (nm) | Measured | Calculated | | Simulated | |
|---|---|---|---|---|---|
| | $\lambda_1$ (μm) | $\lambda_1$ (μm) | $\delta_r$ (%) | $\lambda_1$ (μm) | $\delta_r$ (%) |
| 1200 | 3.1 | 3.5 | −12.9 | 3.5 | −12.9 |
| 2200 | 5.8 | 6.1 | −5.2 | 5.9 | −1.7 |
| 3800 | 11.7 | 11.0 | 6.0 | 9.2 | 21.4 |

## 7. Conclusions

Nanostructured passive scattering arrays composed of dipoles represent a basic structure that makes it possible to study dispersive interactions with electromagnetic waves in the optical domain. A fundamental approach to their design consists of scaling down the dimensions of the elements to achieve selective electromagnetic responses at THz frequencies. However, the specific material parameters of substrates and metallic components have to be considered, since they vary significantly in this frequency domain.

To design the scattering structures, a simple analytical approach can be used. It considers the permittivity of the substrate and the complex dielectric function of the metals in order to evaluate the effective index of refraction applicable to the structure response calculation. A more advanced approach is to use numerical simulation and analysis. However, this can be very demanding in view of the computational time required for complex structures, while also respecting the parameters of the real materials.

In the presented work, we designed scattering dipole arrays intended for operation in the middle infrared band. The design was performed using a simple analytical approach, and also via a numerical analysis approach. In the case of the numerical approach, we examined a simplified numerical model considering only the perfect electrical conductor as the material for metallic part of the structure. Furthermore, a periodic condition was imposed on the model boundaries in order to simulate an array of multiple dipoles.

To lower contamination and to reduce processing time, complementary dipole structures were fabricated using the focused gallium ion beam procedure. Two arrays with defined dipole lengths were successfully produced and further characterized on the basis of their spectroscopic reflectance measurements.

The obtained results showed that the use of even a simple analytical calculation or a simplified numerical simulation can provide meaningful results that, with some exceptions, essentially differ only in percentage units. This is mainly useful prior to structure fabrication. Moreover, the simple numerical model also highlighted the effect of probable basic dipole mode splitting.

More precise results might be obtained by building a numerical model that respects the parameters of the real materials. Additionally, a thinner metallic layer should be used for structure fabrication in order to achieve a uniform FIB milling profile. This combination is planned for further research. Simultaneously, a substrate material with lower losses in the middle infrared band will be considered for sample fabrication.

The main contribution of the present work is related to our proposal of the possibility of using arrays of EM scatterers as miniaturized tags for data encoding. The most important part of this contribution consists in the novel research of a possible design procedure for finding a suitable geometry for the array's elements. Using the EM simulations and experimental measurements, we demonstrated that a simplified analytical approach could be used for the design of the geometry of the array's elements. This will significantly save time compared to designs that rely on full-wave EM simulation. Finally, we demonstrated that FIB milling can be used for the fabrication of EM scatterer arrays in the optical region. This will also save time compared to procedures based on the EBL and lift-off processes.

**Author Contributions:** Conceptualization, P.D. and D.N.; methodology, P.D.; validation, P.D., D.N. and T.K.; investigation, P.D.; resources, P.D.; simulations, T.K. and D.N.; samples characterization P.D. and A.N.; data curation, R.K. and A.N.; writing—original draft preparation, P.D.; writing—review and editing, R.K.; visualization, R.K.; supervision, P.D. All authors have read and agreed to the published version of the manuscript.

**Funding:** This research was funded by Czech Science Foundation, grant number GA15-08803S.

**Institutional Review Board Statement:** Not applicable.

**Informed Consent Statement:** Not applicable.

**Data Availability Statement:** Not applicable.

**Acknowledgments:** Part of the work was carried out with the support of Nanofabrication and Nanocharacterization Core Facility of CEITEC—Central European Institute of Technology under CEITEC—open access project, ID number LM2011020, funded by the Ministry of Education, Youth and Sports of the Czech Republic under the activity "Projects of major infrastructures for research, development and innovations".

**Conflicts of Interest:** The authors declare no conflict of interest. The funders had no role in the design of the study; in the collection, analyses, or interpretation of data; in the writing of the manuscript; or in the decision to publish the results.

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
