# Peer review of "Simulation and Characterization of Nanostructured Electromagnetic Scatterers for Information Encoding"

_electronics, doi:10.3390/electronics11203283_

Round 1

Reviewer 1 Report

The paper addresses nanostructured passive scattering arrays, in particular the specific spectral response regarding the dimensions and material properties, which can be used as tuned devices for encoding of passive tags, i.e. a very original version of an optical device with an optical signature operating as an RFID tag.

Analytical and numerical designs are provided, as well as experimentation results from an implementation. It has been shown that the proposed simple analytical model provides relatively accurate results.

The results shown in Fig. 9 are not clear to me. It is indicated that this figure comprises results for dipoles with different lengths. Furthermore, measured and simulation results are indicated in text, but it does not seem to be the case in the figure. Please take into attention this, improving also the axis identification.

The paper needs extensive writing review. Among the following writing issues  (more can still be found):
Missing the verb in: "Possible application of nanosized scattering structures might in the security field."
Rephrase: Novel is a research into the design and application of passive structures comprising an array of resonant scatterers which can be mutually tuned on different fre-
quencies...
To achieve the such effects -> To achieve such effects
Rephrase: The goal of this contribution is to present some of conclusions in the design of passive scattering arrays for infrared optical range of electromagnetic field.
Line 107: remove additional space before the second symbol
Rephrase: While the commonly used metals, as for example gold, experience a significant change in real and imaginary part of their dielectric function.
In one of our previous work -> In one of our previous works
Line 123: can't -> cannot (formal English)
Line 138: doesn't -> does not (formal English)
Line 163: don't -> do not (formal English)
Please revise the writing.

Author Response

Dear reviewer,

thank you for pointing out the language shortcomings present. I removed the ones mentioned and went through the manuscript again to correct the others.

In the case of the mentioned figure 9, I modified the description in the text and the figure itself, and I believe that the description will now be understandable. Thank you for the remark, the description was trully confusing.

Reviewer 2 Report

In this manuscript, authors present a fundamental approach to their design of the nanostructured passive scattering arrays to achieve selective electromagnetic response at THz frequencies. Within the presented work, authors have performed the design of scattering dipole arrays intended for operation in the middle infrared band. The subject of the paper is interesting. However, here are some problems need to be addressed. 1. The English writing should be improved, there are some errors. 2. The Fig.1 is not clear, which should be improved. 3. It should be given some comment about how to design the nanostructured passive scattering array, and the unit-cell should be illustrated detailed clearly. 4. What are the dielectric constant of the supporting substrate of the nanostructured passive scattering array in practical simulation and experiment?

5. More details about simulation should be given, for example, what kind of boundary conditions, what kind of wave source.

6. For discussing the nanostructurs, the authors should reference other relative articles published earlier: [A] Langmuir 2020, 36, 2, 600-608; [B] Optik - International Journal for Light and Electron Optics 229 (2021) 166300.

Author Response

Dear reviewer,

thank you for your comments. I went through the manuscript for the team of authors to correct language deficiencies.

We have improved the comment on Figure 1 and we believe it is now understandable.

Regarding your third remark, we have included a comment in the Introduction that explains how a set of nanostructured passive scattering arrays can generally be designed to encode data: “Each array will be tuned on defined frequency which provides the data encoding. Typically, two selected tuning frequencies might be used for the binary information encoding. Then a set of 10 arrays would store a 10-bit information.”

The dielectric constants and functions of the metal and the substrate, which were used for the analytical calculation and for the numerical simulation, are in Table 1. For better clarity, we have also added a reference to the table in the text.

Reviewer 3 Report

The authors presented Simulation and Characterization of Nanostructured Electro- 2 magnetic Scatterers for Information Encoding. It was well presnted with good analytical and experimental analysis. The work has significant scientific value. This reviewer recommend to accept the article after addressing the following comment:

1. Please do re-write the abstract. The abstract does not reflect the overall value and contribution of the work. I would recommend to re-write the abstract in manner that the abstract should begin with a brief but precise statement of the problem or issue, followed by a description of the research method and design, the major findings, and the conclusions reached.

Author Response

Dear reviewr.

We rewrote the text of the abstract according to your suggestion and we believe that it is now more clear.

With best regards,

Team of authors

Reviewer 4 Report

The authors present Simulation and Characterization of Nanostructured Electromagnetic Scatterers for Information Encoding.

The paper is well written and it is a very interesting topic.

However more information about the simulation model is expected. A real model for simulation, in my opinion, can be easily implemented using commercial software such as Ansys or CST among others, for example.

I don't know if the comparison with chipless RFID is a good comparison, I suggest to the authors to see Frequency Selective Surface (FSS) in THz band, or tunable FSS in the same range of frequency. 

The analysis must be improved regarding the main resonance and side resonance (similar phenomenon can be found in recent literature).

Up-to-date and weighty references are expected.

Finally, the contribution is not clear and must be highlighted.

Author Response

Dear reviewer,

thank you for your valuable comments to our manuscript. According to your recommendation we provided more details to performed simulations of the studied structures.

As you mentioned, the research field of frequency selective surfaces (FSS) is very similar. However, in the case of FSS all the elements of such structures are generally designed to work on given unified frequency to achieve a common functional frequency response. On the other side, when we will consider a set of arrays with two different resonant frequencies, we can provide binary data encoding. Certainly, the FSS problematics should be mentioned. Therefore, we added a comment on this into the text and we also provided recent references on the works in recognized journals. Thank you for notifying us on this shortcoming of the manuscript.

We also added more emphasized description of contribution at the end of the Conclusion chapter.

Thank you for your comments and recommendations.

With best regards,

Team of authors

Round 2

Reviewer 1 Report

The authors have improved the quality of presentation of the submitted paper,  now easier to understand and follow the ideas. It seems now at a suitable phase for publication.

Reviewer 2 Report

I think that the current version can be accepted

Reviewer 4 Report

All my comments were attended. The manuscript was improved. The paper is now, in my opinion, suitable for publication.